# A Method for Risk Assessment Evaluating the Safety, Stability and Efficacy in Clinical Practice of Anticancer Drug Preparations in the Centralized Compounding Unit of the Veneto Institute of Oncology-IRCCS

**DOI:** 10.3390/pharmaceutics15051429

**Published:** 2023-05-07

**Authors:** Nicoletta Rigamonti, Jessica Sebellin, Francesca Pipitone, Nicola Realdon, Debora Carpanese, Marina Coppola

**Affiliations:** 1Pharmacy Unit, Veneto Institute of Oncology (IOV), Istituto di Ricovero e Cura a Carattere Scientifico (IRCCS), Comprehensive Cancer Centre, 35128 Padua, Italy; 2Pharmaceutical and Pharmacological Sciences Department, University of Padua, 35122 Padua, Italy; 3Immunology and Molecular Oncology Unit, Veneto Institute of Oncology (IOV), Istituto di Ricovero e Cura a Carattere Scientifico (IRCCS), Comprehensive Cancer Centre, 35128 Padua, Italy

**Keywords:** chemotherapy compounding, risk assessment, quality assurance system, risk-based predictive extended stability

## Abstract

Background. Preparation of injectable anticancer drugs in hospital pharmacies is a high-risk activity that requires a proper risk assessment (RA) and quality assurance system (QAS) to ensure both a decrease in risk associated with chemotherapy compounding and high quality of the final product, especially in terms of its microbiological stability. Methods. At the centralized compounding unit (UFA) of the Italian Hospital IOV-IRCCS, a quick and deductive method was applied to evaluate the “added value” provided by each prescribed preparation, and its RA was calculated applying a formula that integrates different pharmacological, technological and organizational aspects. According to specific RA range values, the preparations were divided into different risk levels, in order to determine the QAS to be adopted, according to the Italian Ministry of Health guidelines, whose adherence was meticulously evaluated through a specific self-assessment procedure. A review of the scientific literature was carried out to integrate the risk-based predictive extended stability (RBPES) of drugs with data concerning their physiochemical and biological stability. Results. Based on the self-assessment comprising all microbiological validations of the working area, personnel and products, the microbiological risk level within the IOV-IRCCS’ UFA was defined through the creation of a transcoding matrix, conferring a microbiological stability to preparations and vial leftovers of a maximum of 7 days. The calculated RBPES were successfully integrated with stability data from the literature, leading to the drafting of a stability table of drugs and preparations in use in our UFA. Conclusions. Our methods allowed us to perform an in-depth analysis of the highly specific and technical process of anticancer drug compounding in our UFA, ensuring a certain grade of quality and safety to preparations, especially in terms of microbiological stability. The resulting RBPES table represents an invaluable tool with positive repercussions at organizational and economic levels.

## 1. Introduction

Anticancer drugs are nowadays used in composite and individualized treatment protocols, with the aim of implementing more personalized medicine. Overall, the chemotherapeutic process is complex, and errors can occur at all stages, ranging from prescription and compounding to administration. All these stages must be carefully monitored in order to contain and reduce the risk of errors that can impact the health of patients and healthcare personnel involved [1,2,3]. To contain the risk, the compounding activity takes place in dedicated centralized production units (CPU) in hospitals, named UFA (Unità Farmaci Antiblastici, unit for cytotoxic drug preparations) in Italy [4], under the responsibility of the pharmacists whose competences include organization, management and chemotherapy validations, while technicians are responsible for the final preparation [5,6]. The centralization encourages the standardization of processes, and guarantees a higher quality of care, contemporarily avoiding medication errors that can be fatal in the oncology ward [4]. In addition, centralization reduces production waste, is less time-consuming, and exposes fewer operators to chemotherapeutic drugs during the preparation process [7,8]. Thus, this organization proved to be of paramount importance to ensure the quality and safety of chemotherapeutic preparations, which directly influence their efficacy and limit the risk of iatrogenic toxicity. This is particularly relevant for the parenteral-administered anticancer drugs, which are considered high-alert medications due to their specific characteristics and necessities. Indeed, they are sterile products that must be prepared under aseptic methods. They are prescribed following validated schedules with personalized calculated doses based on patient’s gender, age, weight and/or body surface area, and are constantly updated according to the patient’s global health status. Additionally, most anticancer agents have a steep dose–response relationship and a narrow therapeutic index. Overall, it is evident that chemotherapy compounding is a risky business [9,10]. Compounding of extemporaneous preparations in pharmacy is completely different in terms of risk analysis compared to the batch production in the pharmaceutical industry. Therefore, the Good Manufacturing Practices (GMP) applied to industrial products cannot fit in this context, where preparations are for immediate use, and the variables affecting the quality and safety of the final product are more individual, and related to technical, organizational and structural aspects. Therefore, to lower the risk and contemporarily ensure the highest quality of the final product, each CPU/UFA should adopt dedicated national/international laws, tailoring and optimizing them according to their own working context, designing and applying an appropriate quality assurance system (QAS), to be periodically checked through a self-assessment procedure. However, before preparation, the pharmacist and the prescriber should always consider the risks for the patient and compounding personnel, which include an analysis of the real benefits brought by the preparation (namely the “added value” of the preparation) and the QAS applied to its production, versus the risks related to the unavailability of this medicinal product [11]. To this aim, the Italian Society of Compounding Pharmacists (SIFAP) and the Italian Hospital Pharmacy Society (SIFO) released a position paper to guide UFAs in the above-mentioned processes, especially for evaluation of the added value of each preparation (analysis of risks versus benefits), as well as suggesting a model procedure for the risk assessment (RA) based on the European Resolution CM/Res (2016) [11]. An important issue strictly related to the risk of chemotherapeutic drug compounding, is the restricted shelf-life of the majority of active ingredients (a.i.) and derived preparation, according to their Summary of Product Characteristics (SPC), which for the majority of anticancer drugs is extremely restricted (from hours to 1-2 days). The limited stability not only negatively impacts on hospitals from an economic and organizational point of view, but also on healthcare personnel exposure and patient’s management. Notably, this restriction is only precautional to avoid microbiological contamination. Indeed, several studies have shown that widely used anticancer preparations are much more stable if prepared under validated aseptic conditions, which is the case in CPUs/UFAs [12,13,14,15,16,17]. Therefore, it is reasonable to state that a CPU/UFA is able to guarantee the microbiological stability of its preparations in a precise timeframe, depending on the validated QAS. Consequently, when solid scientific data are available, it is possible to integrate the calculated microbiological stability with data regarding the physiochemical (and biological) stability of compounded drugs. In this regard, it is possible to draft a risk-based predictive extended stability (RBPES) of anticancer drugs. Hence, this work describes the design and application of a method that integrates national and international laws and SIFAP/SIFO guidelines, for evaluation of the risk associated to anticancer preparations, the relative QAS to be applied, and the resulting prediction of their RBPES, in order to improve the quality and safety of parenteral chemotherapies compounded in an Italian public hospital, the Veneto Institute of Oncology-IRCCS (IOV-ICCS, Padua, Italy). Notably, with appropriate minimum modifications, our method can be adapted to different national and European realities. The IOV-IRCCS center is a public health care institute that carries out cancer prevention, diagnosis and treatment, and at the same time performs preclinical, translational and clinical research; thus, it is recognized by the Italian Ministry of Health as IRCCS (Scientific Institute for Research, Hospitalization and Health Care) and Comprehensive Cancer Center. The IOV-IRCCS’ UFA compounds conventional and experimental chemotherapies for the center and for its branch located in Castelfranco Veneto (Treviso, Italy), and, thanks to specific agreements, for two other public hospitals in Padua.

## 2. Materials and Methods

### 2.1. Determination of the “Added Value” of Antiblastic Preparations

To determine whether each of the preparations prescribed at IOV-IRCCS can be of “added value”, and therefore needs to be prepared in accordance with the national and European legislation, the decision tree designed by SIFAP/SIFO is applied (Figure 1) [18].

### 2.2. Risk Assessment

The risk associated with each preparation (RA) is calculated by adding the pharmacological risk (X) to the product obtained by the technological risk (Y) multiplied by the risk depending on the number of preparations per year (Z), using the formula: RA = X + (Y × Z). X is determined by the intrinsic characteristics of a.i. and by the route of administration, while Y is calculated according to the complexity of the preparation, considering the number of components, calculations and manipulations, as well as the route of administration, according to the formula: Y = A × B × C × D. For each criterion, a score between 1 (less critical) and 5 (more critical) is assigned. As regards the A value, it is possible to subtract from the calculated value 1 unit up to the minimum value of 4 in the case of compounding of anticancer drugs and parenteral nutrition bags in which the calculations are carried out with specific management software. Z is calculated based on the number of preparations of the previous year (Figure 2) [18]. The calculated RA is categorized in tertiles.

### 2.3. Design and Application of the QAS

Chemotherapeutic drugs for parenteral use can be considered high-risk preparations. Therefore, the QAS was designed in the light of the Italian Ministry of Health guidelines, with particular focus on the Norms of Good Preparation (NBP) [19] and Annex 1: “Manufacture of sterile products” (Good Manufacturing Practice, GMP—European Commission) [20], and quality controls on the working area and preparations were delineated and periodically programmed. Drug compounding at IOV-IRCCS’ UFA is carried out in a Grade A cleanroom (ISO 4.8 according to EN/ISO 14644-1) by means of a unidirectional vertical laminar airflow hood with HEPA filter, bio-safe BIOHAZARD, Class II, according to EN 124644-1. This hood is inserted inside a Grade C cleanroom. There are filter zones to access the Grade D background room (Figure 3). All medical devices used are closed systems without gas exchange.

Microbiological controls are performed according to the NBP and Annex 1 of the European Union Guide to GMP (EU GGMP). Of note, the environmental microbiological controls are carried out every two weeks, and consist of microbial air and microbial contamination of surface and operator’s gloves. Materials used are swabs (Copan, Brescia, Italy), 90 mm Petri dishes with Sabouraud dextrose agar (SDA, Becton Dickinson, Franklin Lakes, NJ, USA) and 90 mm Petri dishes with trypticase soy agar (TSA, Becton Dickinson). SDA is used for the detection of yeasts and fungi, while TSA is a general growth medium. Surface microbial contamination is performed by means of a swab in the central working area of the hood during the operating phases. Air monitoring in preparation laboratories is carried out using the plate exposure method. Briefly, Petri dishes with SDA and TSA mediums are positioned inside the hood and in the adjacent environment. Sampling is performed during normal compounding conditions. Plates are left for 1 h under the hood and for 4 h in the laboratory environment. Contamination of the operator’s gloves is conducted using Petri dishes with TSA culture medium. Three plates are placed on the worktop of the hood, one for the right glove, one for the left glove and one for validating the result. The fingers are placed for about 10 s on the corresponding plate applying light pressure. Swabs and plates are sent to the Microbiology and Virology Unit of the University Hospital of Padua, to be analyzed for the possible growth of colonies. Results obtained are compared with the limit values for microbiological contamination reported by Annex 1 of the EU GGMP.

The microbiological validation of the process is carried out through the media-fill test, performed in the hood 3 times at initial qualification, and in a single test every 6 months for the requalification. During the execution of the validation test, a microbiological monitoring of the environment is also carried out at the same time, as mentioned above, including sampling the footprint of the gloves of the operator involved. The test principle is to reproduce the process exactly but replacing real products with culture media. Material used for the execution of the test are 2 bottles of Tryptone Soy Broth (TSB, Merck KGaA, Darmstadt, Germany), empty sterile bags (Baxter, Deerfield, IL, USA), syringes (Becton Dickinson), vial spike (IcuMedical, San Clemente, FL, USA), bags spike (IcuMedical). Operators’ gloves media-fill test is performed in triplicate (once a day for 3 days or 3 times in a day) once per year for newly hired staff, and every six months for experienced staff, at once in the middle of the day. The test requires each operator to prepare: a bottle, 10 bags, 5 syringes, a mother bag, 3 satellite bags (with scalar dilutions), plates with fingerprints of the gloves at the end of the work shift, hood/operator control plates and environmental control plates (laboratory). All preparations are then incubated for 7 days at 20–25 °C immediately followed by 30–35 °C for 7 days and regularly observed to detect the appearance of possible disturbance indicating microbial growth. In case of positive result, the microorganisms are determined as well as the step of the process at the basis of the contamination.

Airborne particulate control is carried out every 6 months, both during the execution of the process validation test and of the environmental control, using the Fluke985 (Everett, WA, USA) particle counter instrument, to measure the efficiency of the filters and carry out an assessment of the air quality. The limits of airborne particulate concentration considered are those reported by Annex 1 of the EU GGMP. Notably, the scheduling of media-fill tests on each operator, airborne particle and environmental microbiological tests is defined on the basis of the annual number of preparations set up in the UFA (approximately 200,000/year).

### 2.4. Self-Assessment of the Adherence to the QAS

The self-assessment considers all of the stages concerning drug compounding (Table 1), for which a score is given based on adherence to the QAS: N.A. (not applicable); 1 for total absence of adherence; 2 for ≤50% adherence, which means that there is documented evidence of ongoing organizational settings or tests for the requisite. A score of 3 is for >50% adherence, as the requisite is close to being fully delineated or settled, or completely but not systemically realized and, therefore, procedures and workflow are well established, but complete documentation is not available. A score of 4 is for complete (100%) adherence to all the procedures of the QAS, which is systemically settled and there is complete documentation attesting its application and monitoring.

As proof of the quality of the entire procedure, we selected some of the most prescribed a.i. and derived preparations at IOV-IRCCS. For each selected a.i., vial leftovers were used as such or employed for the preparation of additional infusion bags (simultaneously compounded with the prescribed ones), to be stored for 7 days, under the conditions required by the relative RCP. The microbiological contamination of these samples (pool of three batches) was evaluated after incubation at 25 °C and 37 °C for 14 days in TSB or fluid thioglycolate medium (FTM, both from Merck KGaA). The presence of pyrogens was assessed for each single sample using the Limulus Amebocyte Lysate (LAL) PYROGENT™ Plus Gel Clot test kit with 0.125 endotoxin unit (EU)/mL sensitivity (Lonza, Basel, Switzerland), according to the manufacturer’s instructions.

### 2.5. Literature Review

A review process of the physiochemical and biological stability of each a.i. was performed, and regularly reviewed and updated, in order to draft a RBPES table based on both the updated SPCs and the currently available scientific literature. The bibliographic research was carried out using Stabilis^®^ and Micromedex databases, as well as PubMed. Specifically, the evaluation was primarily based on the data reported in the relative SPC. Then, Stabilis^®^ and Micromedex databases were consulted for the concentration ranges of each preparation in use at IOV-IRCCS, selecting only studies with a high level of evidence (mainly level A+, A). In addition, the excipients were compared between the various SPCs provided for the same molecule. Then, to implement the extracted data, PubMed was performed, using the name of the a.i., vehicle for dilution/resuspension and type of container. No time limits were set. Only studies with medium-high compliance with the European consensus published by Bardin et al. [21] were taken into account.

## 3. Results

### 3.1. RA and Selection of the Relative QAS

The numerical value obtained from application of the formula RA = X + (Y × Z), determines the quality system to be adopted for the specific preparation. Toward this aim, a transcoding matrix was created that is categorized according to RA tertiles (Table 2).

For example, the RA for a bag of paclitaxel (80 mg/mq) diluted in 0.9% sodium chloride injection can be outlined as follows. The preparation has an added value since there is evidence of the absence of an industrial product with the same qualitative/quantitative composition, pharmaceutical form, dosage and excipients. The a.i. is reported in monographs of both Italian and European Pharmacopoeias. Since it is a chemotherapeutic drug for parenteral administration, X = 125. Assuming the patient has a body surface area of 1.6 mq, the dose to be administered is 128 mg in a final volume of 250 mL of 0.9% sodium chloride injection. The initial value of A = 5. However, since compounding of anticancer drugs in our UFA is carried out by aid of specific management software, A = 4. B = 1 since the preparation is a solution, C = 5 for sterile parenteral solutions, D = 5 for the sterility of the entire process. Therefore, Y = 100. Z = 1.2 since at IOV-IRCCS more than 500 preparations were carried out in 2022. Overall, RA = 245. However, for a laboratory certified according to a recognized QAS and whose certification is correctly maintained and verified according to the frequency provided by the accreditation system, the RA value can be lowered by 30%, and if the personnel have experience of >5 years, RA can be further lowered by 2.5%. Therefore, RA = 167.2 and a medium risk is assigned to the preparation.

### 3.2. QAS Application and Control

The results of the microbiological controls performed in 2022 at the IOV-IRCCS’ UFA are reported in Figure 4, together with the reference limit values. Tests that detected microbial contamination, even if below the agreed reference limits, are highlighted in light orange.

As a proof of the aseptic quality of the entire process, vial leftovers and compounded products in infusion bags proved to be negative for microbial growth according to LAL test and after 14 days of incubation in TSB and FTM mediums (Figure 5).

### 3.3. Prediction of the Microbiological Stability of Compounded Medications

Based on the degree of microbiological contamination of the environment and process validation, the microbiological risk level within the UFA can be defined, which is essential for defining the maximum microbiological stability of each preparation. Therefore, based on the type of controls and environmental classification of the working area, it is possible to define a risk level that is able to ensure a high degree of quality and microbiological stability to the preparation. In this regard, a transcoding matrix was outlined indicating the possible cases and the types of medical devices (MD) in use at the compounding centralized unit, and the relative predicted microbiological stability of the drug (Figure 6). Based on the characteristics of our UFA, together with the negative results of the tests carried out, the microbiological stability of compounded drugs and opened vials can be assigned to a maximum of 7 days. 

### 3.4. Draft of the Stability Table of a.i. and Relative Preparations Compounded at IOV-IRCCS

Once validated a prolonged microbiological stability, it is important to also determine the physiochemical and biological stability of the drug, in both the original opened vial and its derived preparations, considering all vehicles, excipients and the containers or packages employed. A systematic and critical review of the literature was performed. In the final choice of the stability to be applied to each preparation, based on the guarantees assured from the QAS applied, the 7 day threshold was never exceeded (with the only exception for Blincyto) and, where the literature presented contradictions, the conservation limits have always been largely restrictive and precautionary. For almost all molecules, a stability at both room temperature and 2–8 °C was defined, and when studies proved low stability of a.i. at low temperatures, a stability of 24 h at 2–8 °C was considered. On this ground, a summary table of a.i. and related preparations in use at the IOV-IRCCS (Figure 7) was drafted, which reports: description of the a.i., brand name, method of reconstitution (volume and solvent to be used), concentration of the original solution, volume of dilution using sodium chloride 0.9% or glucose 5% solutions, stability of the opened vial leftover, use of 0.2–0.22 µm filter, photosensitivity of the drug, time range of infusion of the compounded drug, stability of the diluted drug according to selected literature and to the SPC, and eventual notes for compounding. Notably, with this table, an extension of the stability, with respect to the SPCs, of expensive molecules such as trastuzumab, bevacizumab, atezolizumab and rituximab was assigned. From an organizational point of view, in order to avoid excessive variability of the dilution volumes during the preparation, a simplification of dilution ranges was performed. In accordance with the concentration ranges of stability, working at “increased volume”, which means inserting the required volume of the drug into the bags without removing an equal volume of diluent, turned out to be compatible for the majority of the drugs, reducing the number of operations required for compounding, and consequently the overall risk.

## 4. Discussion

The chemotherapeutic process is risky for both operators and patients, with error subject to occurring at all steps, ranging from prescription and compounding to administration. The constant expansion in the number of these preparations and their use in complex treatment protocols have led to an increase in the likelihood of error, especially in the compounding activity, which is extremely critical in the case of anticancer drugs for parenteral use [10]. The variables influencing the quality, efficacy and safety of these products are manifold, and can be divided into three major categories: technical, organizational and structural [8,9]. On the other hand, the risk value associated with anticancer drug preparations is directly dependent on the QAS applied [1,22].

After demonstrating the added value of each preparation through the application of the decision tree [18], it is necessary to evaluate the three types of risk: pharmacological and technological risks, and the volume of activity. In this regard, it is mandatory to design and apply an appropriate QAS to the specific compounding activity, which is based on identification of suitable qualification systems for critical factors and process validation. Importantly, a periodic self-assessment must be performed as it is essential to pinpoint the grade of adherence of the working context to the requirements of the NBP/GMP, and additionally it serves as a useful method for quality improvement [4].

A sterile preparation set up in a centralized unit is considered stable for a given extent of time when, under defined conditions of temperature, humidity and exposure to light and suitable packaging, its essential properties do not change or change within tolerable limits. This is of the utmost importance for chemotherapeutic drugs, especially considering the difficulties of planning drug compounding in advance. Notably, if the environmental standards of the laboratories where drug compounding occurs is able to guarantee the absence of microbiological contamination of the preparation, it is possible to also ensure the extent of their period of validity [22,23,24,25]. Indeed, the narrow stability limits conferred by the pharmaceutical industry to the majority of a.i. and related preparations are principally based on the possible risk of biological contamination, and do not rely on their real physiochemical stability. However, when compounding medicines for patients occurs in centralized units, several aspects are of paramount importance and therefore are strictly controlled: dose accuracy, sterility assurance, occupational exposure, and stability under clinical practice conditions [21]. On this ground, here, we demonstrated that, when the sterile conditions during the manufacturing process and maintenance of the sterility of the final product are ensured through proper application of RA and QAS, it is possible to precisely calculate the microbiological stability timeframe of the compounded products. However, our future planning is to create a standardized protocol for periodic analysis of the microbiological content and stability of sample batches of compounded drugs, as it is already performed for working areas and personnel.

Once the microbiological stability of preparations in a centralized compounded unit are validated, the pending issue remains validation of their physiochemical stability and, for preparations containing biotechnological a.i., maintenance of their biological activity. To this aim, extensive research through the major databases for experimental data regarding the physiochemical and biological stability of a.i. and derived preparations was performed. As a result, a RBPES table of reconstituted and/or diluted drugs was drafted, considering multiple aspects such as the type of diluent used, the concentration and storage conditions. This table represents an invaluable tool as it paves the way to improve the organization of centralized units and planning of preparations, with a consequent reduction of the clinical risk and exposure risk of the operators [26]. In particular, it contributes to containing economic waste, as it permits the optimal use of vial leftovers, multi-day vial sharing and dose banding procedures, as well as optimization of processing times and human resources [21,27,28]. This becomes even more important in the case of expensive biological drugs such as monoclonal antibodies, or for the setup of therapies with prolonged infusion time using elastomeric pumps. Notably, from our analysis it emerges that stability data for many anticancer preparations is still lacking. Indeed, validation of the physiochemical and biological stability of these drugs is considered a paramount “scientific mission” for the hospital pharmacist [29,30,31]. On this ground, the European Medicines Agency (EMA) published scientific guidelines on human medicines that are harmonized by the International Council for Harmonization of Technical Requirements for Registration of Pharmaceuticals for Human Use (ICH) regarding the evaluation of stability data (ICH Q1A), the stability testing of new drug substances and products (ICH Q1A(R2)), the tests for the validation of analytical procedures (ICH Q2(R1)), the photostability testing of new drug substances and products (ICH Q1B), and the stability testing of biotechnological/biological products (Q5C) [32]. It is a challenging venture, especially because these studies require different and many methodologies, equipment, as well as expertise and that are difficult to gather in a single institution [8,33,34,35]. Taking advantage of the close connection between preclinical and clinical activity at IOV-IRCCS, its equipment and the presence of specialists with heterogeneous expertise such as pharmacists, chemists, biologists and technicians, we are setting up a joint working group, envisaging to validate the long-term stability of high-cost drugs, especially mAbs and their biosimilars [36], with positive impact on healthcare costs, patient compliance and managing, and exposure safety of the healthcare personnel.

## 5. Conclusions

Our study reports a quick and deductive method to understand and ameliorate the compounding process in a hospital centralized compounding unit, especially in terms of organization, management and chemotherapy validations, as well as assurance of the safety and quality of the final product. To our knowledge, this is the first documented application of suggested guidelines of SIFAP/SIFO integrated with national and international laws. Additionally, we demonstrated the feasibility of extending the stability of compounded chemotherapies and vial leftovers by integrating the calculated RBPES with high-evidence scientific data of the physiochemical and biological stability of each anticancer drug, as well as creating a widely applicable tool, represented by the summarizing table of characteristics and extended stability of compounded anticancer drugs in use in hospitals, supporting the optimization of the chemotherapeutic process at different levels.

## Figures and Tables

**Figure 1 pharmaceutics-15-01429-f001:**
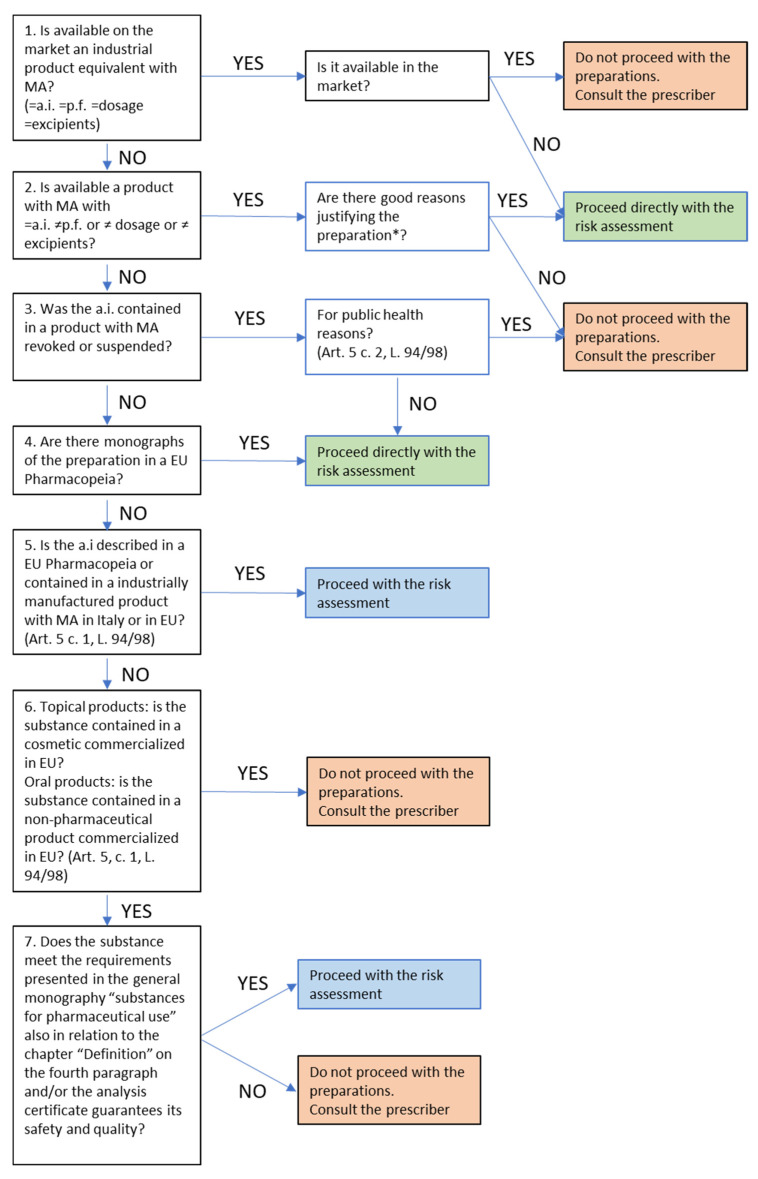
Decision tree for drug compounding. MA = Marketing authorization; EU = European Union; a.i. = active ingredient; p.f. = pharmaceutical form; Art. 5, c. 1, L. 94/98 and Art. 5 c. 2, L. 94/98 = Italian Legislative Decree 94/1998, Article 5, paragraph 1 and Italian Legislative Decree 94/1998, Article 5, paragraph 2, respectively. * Intolerance, allergy, idiosyncrasy to an excipient. In this case: use capsule shell of plant origin; improve the taste; make it easily washable by adding surfactants of semi-solid preparations for cutaneous use; do not use fragrances or dyes; change the pharmaceutical form from a capsule to a gel to be dissolved (pediatric patients, effects of a stroke or Alzheimer’s disease); set up a preparation for a substance with a low therapeutic index.

**Figure 2 pharmaceutics-15-01429-f002:**
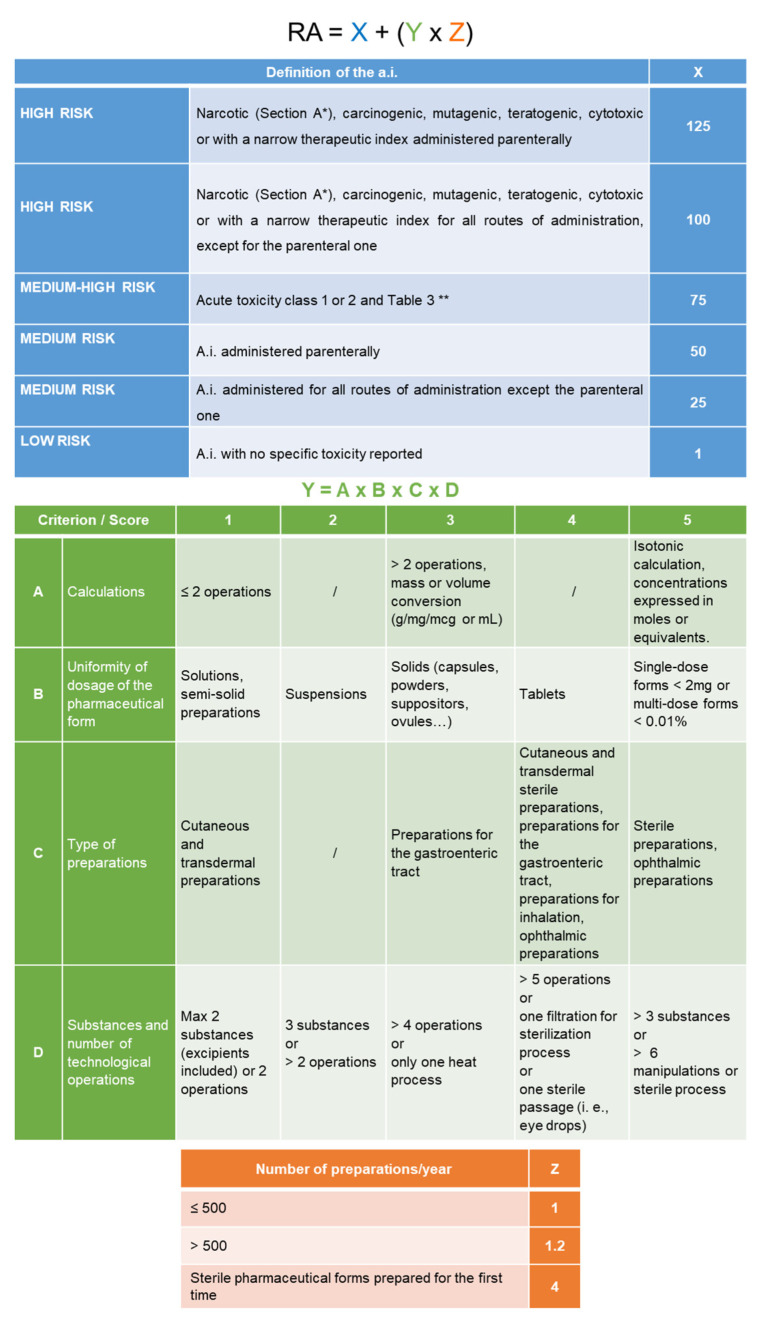
Calculation of RA and score assignment (modified from [18]). For each preparation, RA is calculated based on the score of the pharmacological risk (X, blue table) to the product obtained by the technological risk (Y, green table) multiplied by the risk depending on the number of preparations per year (Z, orange table). * Refer to Table of Medicines V, Section A (art. 14 DPR 309/90, mod. D.L. 36/2014); ** Refer to Table 3 of Substances to be Kept in a Secure Locker [19].

**Figure 3 pharmaceutics-15-01429-f003:**
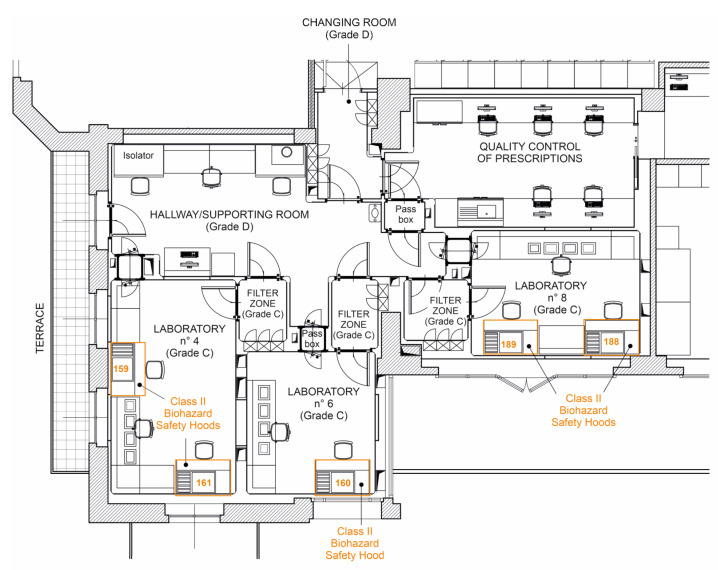
Planimetry of IOV-IRCCS’ UFA. Drug compounding is performed in a Grade A cleanroom, according to ISO 14644-1 standards. Therefore, the UFA consists of three Grade C cleanrooms (laboratories number 4, 5 and 6), which have unidirectional vertical laminar airflow hoods with HEPA filters, bio-safe BIOHAZARD, Class II. Orange numbers refer to an internal numbering of hoods. Filter zones permit access to the Grade D backroom.

**Figure 4 pharmaceutics-15-01429-f004:**
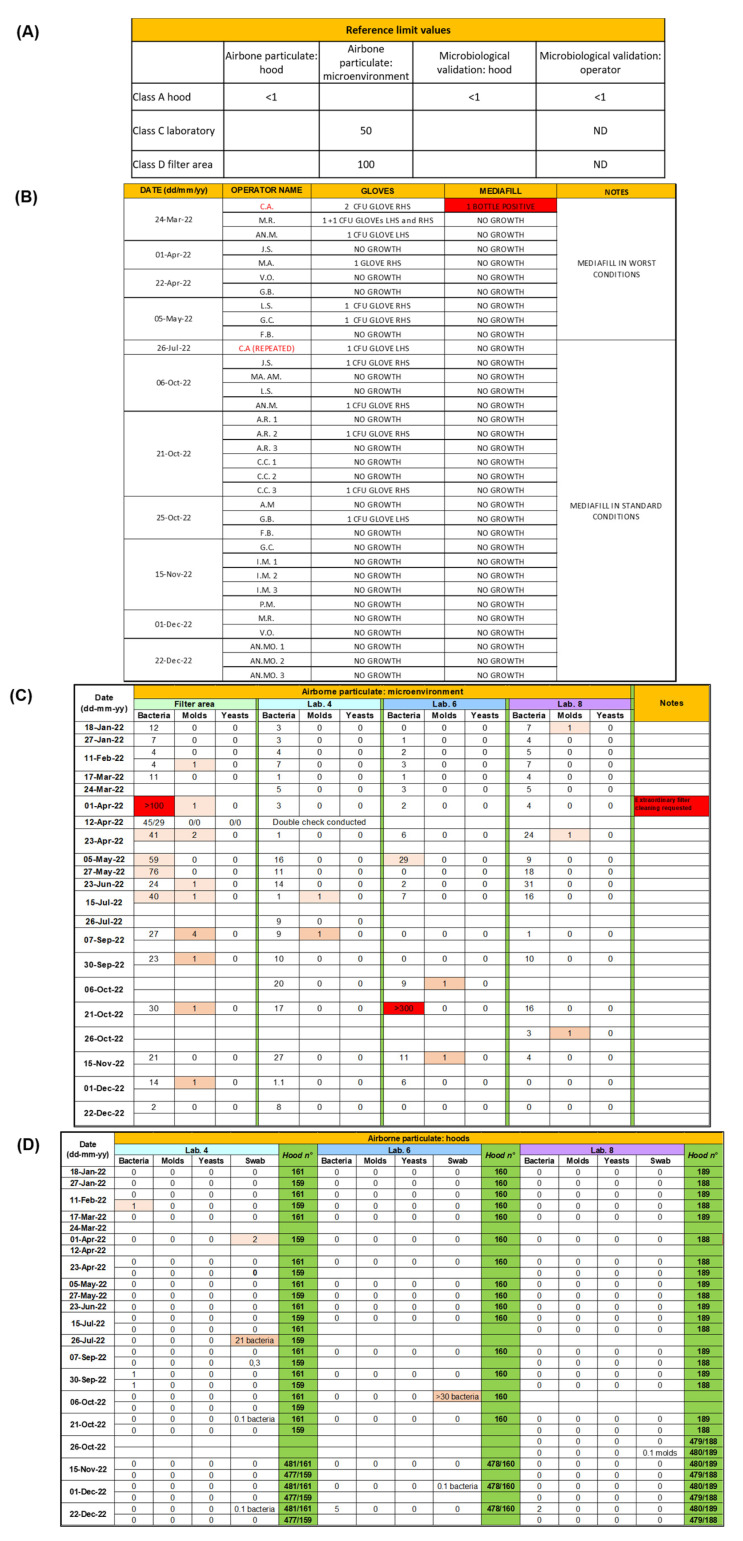
Microbiological controls performed at IOV-IRCCS’ UFA in 2022. The numbers refer to the colony-forming unit (CFU) calculated. (**A**) Reference limit values according to Annex 1: “Manufacture of sterile products” (Good Manufacturing Practice, GMP—European Commission) [20]. (**B**) Operators’ gloves media-fill tests. (**C**) Surface microbial contamination assessment performed using swab in the central working area of the hood during the operating phases. (**D**) Air monitoring in preparation laboratories.

**Figure 5 pharmaceutics-15-01429-f005:**
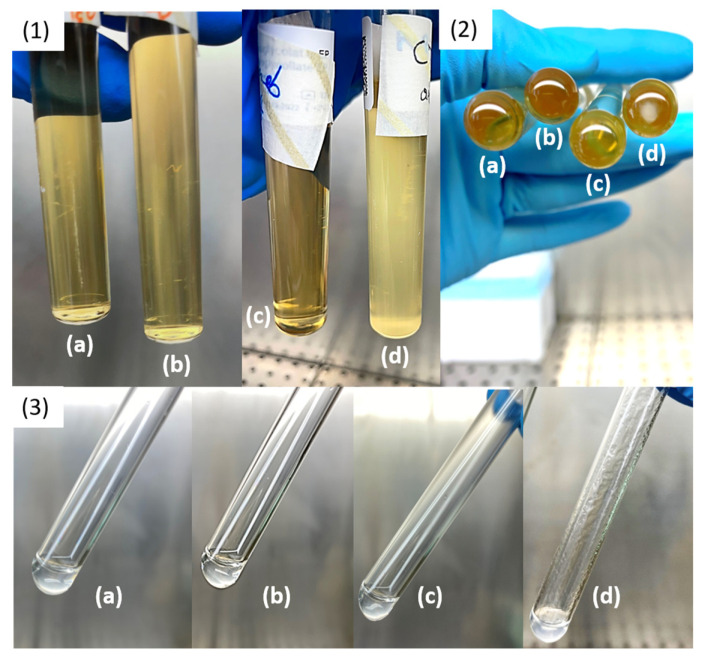
Microbiological tests. Microbiological tests were performed on pooled samples from 3 different batches of pembrolizumab bags diluted at 2 mg/mL (**a**) or 4 mg/mL (**b**) in saline solution, or paclitaxel vial leftovers (**c**), with the relative positive controls (**d**). Samples were incubated in FTM (**1**) or TSB (**2**) for the evaluation of anaerobic and aerobic bacterial contaminations, respectively. The presence of pyrogens was investigated by LAL test (**3**).

**Figure 6 pharmaceutics-15-01429-f006:**
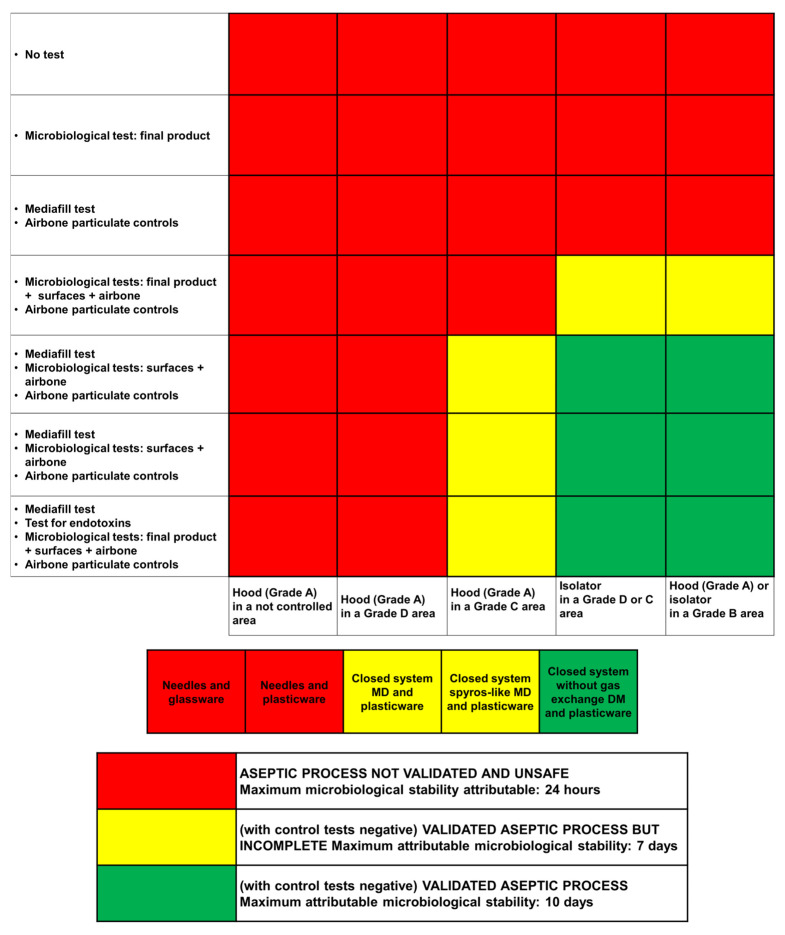
Prediction of the microbiological stability of drugs based on the self-assessment of the QAS. A transcoding matrix for the self-assessment and type of MD used at IOV-IRCCS is generated that permits determination of the maximum microbiological shelf-life of each preparation.

**Figure 7 pharmaceutics-15-01429-f007:**
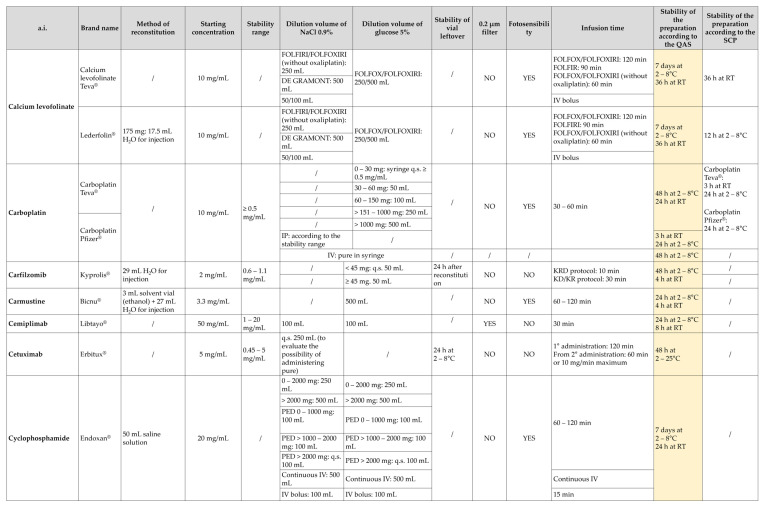
Extracted page of the stability table of the formulations and preparations in use at IOV-IRCCS. The column in light yellow shows the revised stability according to the microbiological stability assured by the QAS and the scientific literature supporting an extended stability from a physiochemical and biological point of view. The table comprises 119 a.i., divided into antiblastics (109), antiretrovirals (2), antidotes (2) and pain relievers (6). Abbreviations: q.s. = a sufficient quantity; IV = intravenous administration; IP = intraperitoneal administration; PED = for pediatric use.

**Table 1 pharmaceutics-15-01429-t001:** Self-assessment of the QAS of the IOV-IRCCS’ UFA, designed based on the Italian NMB and Annex 1 of the EU GGMP as reference texts [19,20].

Critical Factors and Processes	Activity	Requirements According to the QAS for the Self-Assessment
(1) Quality management	Responsibility	The pharmacy director is the general manager. He/she defines the objectives and the quality policy of the pharmacy, assigns responsibilities for critical activities, ensures availability of the necessary resources to maintain the established quality level, and periodically reviews the system to ensure that the objectives are properly defined and efficiently achieved.
	Planning	Planning of activities is a function of quality objectives. The effectiveness of planning is directly related to the efficiency of responding to requests which cannot be easily programmed. The pharmacy must establish its own rules of conduct for the laboratory and validate the processes of drug compounding. The validation of a process is a documented program that gives a high level of confidence on assuming that the process will consistently produce a result that conforms to predetermined specifications and to quality attributes.
	Documentation	Documentation of activities: written procedures are mandatory. Every activity directly or indirectly related to each pharmaceutical preparation must be fully documented.
(2) Personnel	Responsibility and documentation	All personnel must be highly qualified. The person in charge of each preparation is a pharmacist who can delegate the preparation to technical staff or trainees, but always under his/her supervision as he/she holds full responsibility for the entire process. Tasks and responsibilities must be clearly assigned and documented. Detailed programs must be established that provide instructions to each specific task assigned to the laboratory staff, with special focus on the personnel whose activity has a direct impact on the quality of the final product. The quality of medicines prepared in pharmacies derives from the ability and specific competence of the pharmacist in charge who, therefore, should be encouraged to deepen and update his/her own knowledge by attending specific courses and seminars, and consulting scientific texts and technical publications, as well as asking for advice from experienced colleagues.
(3) Laboratory	Qualification	The working area must be separated from other areas and classified according to the RA calculated in compliance with Annex 1 of the EU GGMP. Preparations at high microbiological risk must be carried out in a laminar flow working area of Grade A. The immediate surrounding area must be of Grade B. All preparations deriving from mixing, dilution and partitioning can be set up in a Grade A laminar flow working area, located in an area equipped with a filter zone for particle and microbiological control of the air. Dangerous preparations (e.g., toxic preparations, radiopharmaceuticals) must be handled in special and dedicated biological safety hoods. Entrance to the working area is allowed only through special filter or changing rooms of the same grade. The walls, ceilings and floors must be intact, with rounded corners, easily washable and disinfectable. Rooms must be equipped with an air conditioning system, ventilation and air filtration using HEPA filters (subjected to periodic maintenance and adequate alarm systems), with a number of air changes per hour adequate for the size of the room and activities. All the areas must be in overpressure with respect to lower class areas, except for the handling of cytotoxic substances. The qualification of the working areas is performed by environmental classification of the particle counts at rest and in operation, as well as control on environmental microbiological contamination (air and surfaces).
(4) Equipment	Qualification	All equipment must comply with the current legislation and must satisfy the requirements of the QAS. The laboratory should have groups of electrical continuity that are routinely subject to maintenance. For the compounding of sterile preparations, use of sterile glassware and sterile or sterilized disposables is mandatory, and the sterilization process must be periodically validated. The classification of the working area (hood) includes laminar airflow velocity test (EN 12469), speed test of the front air curtain (EN 12469), test of the direction and display of the air flow.
(5) General activities and operations	Documentation	All procedures and work instructions (including all the controls that must be executed) must be reported in detail in written form, together with worksheets describing all the stages of preparation. Specific detailed procedures must be provided for dangerous and/or harmful products. The instructions must be periodically updated.
(6) Manipulation process in asepsis	Validation	The whole process of aseptic handling must be validated by media-fill test and by continuous monitoring of the microbiological contamination of critical surfaces.
(7) Physiochemical stability of the prepared drug	Validation	Analysis methodology must be properly settled for each preparation, and it must at least comprise visual examination.
(8) Quality control of the final product	Validation	As required by the NBP, the magistral preparations must satisfy the sterility test and the bacterial endotoxin test, if prescribed in monographs. For preparations administered within the time limits defined by a validated system, the sterility test is not required. However, the preparation methods must ensure the asepticity of the product. Therefore, the sterility of the preparation is ensured thanks to the application of a properly validated production process that uses conditions and equipment designed to prevent microbial contamination.
(9) Labelling		The label must comply with current legislation, be clearly legible and indelible, and permanently adhered to the container. The label must at least report: - patient’s name, surname and date of birth; - ward of destination; - qualitative and quantitative composition (active ingredient and dosage); - final volume, expected infusion time, expiration date and storage conditions until use; - name, address and telephone number of the pharmacy (as required by “Recommendation 14”); - date and time of preparation.
(10) Transportation process	Validation	Validation of the transport process via adoption of a continuous monitoring system of the temperature, consisting of a temperature recorder with an internal sensor to verify that it does not exceed 25 °C or drop below 2 °C. The temperature recorder has an accuracy ± 0.5 °C and resolution of 0.5 °C programmed for making 30 recordings at 1 min intervals.

**Table 2 pharmaceutics-15-01429-t002:** QAS level to be applied according to the calculated RA.

Risk Value (RA)	Risk Index	QAS
RA ≤ 50	Low	NBP with minimal required procedures.
50 < RA ≤ 175	Medium	Complete NBP, with specific procedure and periodic quality control of the pharmaceutical form (validated procedure).
RA > 175	High	Complete NBP, with specific procedure and quality controls scheduled with a predefined frequency according to the method and the preparation.

## Data Availability

The data that support the findings of this study are available on request from the corresponding author.

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
