# Peer review of "A Method for Risk Assessment Evaluating the Safety, Stability and Efficacy in Clinical Practice of Anticancer Drug Preparations in the Centralized Compounding Unit of the Veneto Institute of Oncology-IRCCS"

_pharmaceutics, 2023, doi:10.3390/pharmaceutics15051429_

Round 1

Reviewer 1 Report

Dear Authors,

The topic is very important and the idea of the paper is well-founded, but the presentation is very confused and complicated. The manuscript has to be substantially revised - a major revision is recommended.

The manuscript is very complex and contains a lot of data, but it is difficult to follow because it does not have a classic and clear structure of a scientific paper.

The introductory part is too long and should be shortened to state the most important facts; it is necessary to reduce the amount of description and better define the purpose and goals of the research. The final part of the introduction provides a description of the institution, IOV-IRCCS center, and it would be better if the research problem was clearly stated in that part, and the guidelines for its solution were given.

The Materials and methods are not clearly presented. The whole part regarding the experimental design is very difficult to understand for the reader who has no knowledge of this issue. In the description of Figure 1, there are no explanations of all the abbreviations that appear, nor are the abbreviations UE and EU unambiguously used. When mentioning the formula for RA calculation, some reference according to which the formula was deduced has to be cited.

Part of the text from lines 132 to 137 is not clear. A description of the work area is given, which would be much clearer if there was a scheme.

When talking about microbiological controls (line 137), there should be a more detailed description of the procedures, sampling, number of samples taken, references should be cited for the tests that are mentioned, and references should be cited for the reagents used to perform the tests. The same applies to the equipment, for instance Fluke985 particle counter instrument, a better description should be given there.

Table 1 contains a lot of data, but there is no reference to the literature according to which it was compiled.

In line 168, “vial leftovers” are mentioned. It is not clear how many of these samples there were. Also, you mentioned testing with the LAL test kit, but it is not clear what exactly was tested.

In my opinion, it is necessary to completely reorganize the methodological part of the work and clearly define each stage of the research and all methods.

In the Literature review part (lines 179 – 184), the keywords on the basis of which the literature search was conducted have to be defined. It should be also defined what timespan was taken into the search. Then it should be said what criteria were used for the judgment, that is, how it was assessed that you stated "high level of evidence"; based on what was it concluded?

The results are presented in a very complicated manner.

Figure 3 is very difficult to read and understand. Since the areas to which the results refer (Lab 4, Lab 6 and Lab 8) were not defined in the section on Materials and methods, it is difficult to understand the results. Also, the numbers listed in the “hood” columns are difficult to understand. What do they refer to? All this is due to the fact that there is no well-described legend accompanying the table. Appropriate explanations must be given for all of the abbreviations listed in that figure.

In part 3.3. descriptions of the work space are given. These descriptions should be explained in the methodological part of the paper, and the results should only refer to what was found. The present description is confusing and incomprehensible.

In part 3.4. it is stated that a "systematic and critical review of the literature" was made. However, this does not belong to the section Results. All such descriptions should be defined in the chapter on experimental design and methods. Wherever it is stated that something has been done according to the literature, it is necessary to cite in parentheses the appropriate reference numbers on which these statements are based.

Table 6 is too complicated and its content is hard to follow because the font is very small.

The general impression is that the presentation should be made simpler, organized into parts that clearly define what, how and why was studied and with which methods. Figures should be simplified because they are difficult to understand.

Author Response

We sincerely thank the Reviewer for taking time to carefully review the manuscript and give constructive criticisms and valuable comments, which helped to revise and improve the manuscript. Here, the response point-by-point to comments and suggestions.

Point 1. The introductory part is too long and should be shortened to state the most important facts; it is necessary to reduce the amount of description and better define the purpose and goals of the research. The final part of the introduction provides a description of the institution, IOV-IRCCS center, and it would be better if the research problem was clearly stated in that part, and the guidelines for its solution were given.

Thanks to the Reviewer for the comments. The introduction was designed based on a schedule comparable to that used in other works concerning the same topic. We are aware that the topic is complex, as indeed it highlights many issues related to drug compounding in hospitals.

The points are the following:

  1. The management of anticancer therapies in hospitals is an extremely risky activity, especially regarding the compounding step (Lines 38-43).
  2. To lower the risk (for both patients and healthcare professionals), nowadays anticancer drugs are compounded in centralized compounding unit (named UFA in Italy) (Lines 43-55)
  3. A major issue for the centralized compounding units is the management of parenteral anticancer drugs (Lines 55-60)
  4. The legislation for the preparation of pharmaceutical industrial products cannot fit with drug compounding in centralized compounding units/UFAs. Therefore, every unit should adopt such laws, tailoring and optimizing them according to their own working context. To this aim, a quality assurance system (QAS) must be designed, applied and routinely controlled through a self-assessment procedure (Lines 61-73)
  5. The Italian Society of Compounding Pharmacists (SIFAP) and the Italian Hospital Pharmacy Society (SIFO) released a document to guide UFAs in this process, that comprises the evaluation of the real necessity to compound the anticancer preparation prescribed (analysis of benefits versus risks), the importance of calculating the risk associated with that preparation (RA, which strictly depend on both the characteristics of the preparation itself and UFA’s ones), and therefore to design a proper QAS according to the calculated RA (Lines 73-78).
  6. Another important issue strictly related to the risk of anticancer drug compounding is the restricted shelf-life of the majority of active ingredients and derived preparation, according to their Summary of Product Characteristics (SPC) (Lines 78-82)
  7. However, stability restrictions are only precautional to avoid microbiological contamination. Indeed, several studies have shown that widely used anticancer preparations are much more stable if prepared under validated aseptic conditions, which is the case in UFAs. Therefore, an UFA is able to guarantee asepticity, and a consequent microbiological stability of preparations in a timeframe dependent on their QAS. The overall stability of anticancer drugs can be therefore integrated with scientific data ensuring also the physiochemical stability of each a.i. and preparation. In this connection, each hospital can obtain its own risk-based predictive extended stability (RBPES) of anticancer drugs (Lines 82-86)
  8. The IOV-IRCCS is a reference Center in Padua, Italy for anticancer drug compounding. The aim of this study is to apply, integrate and optimize current law and SIFO/SIFAP guidelines to create a proper QAS and obtain a RBPES (Lines 87-100).

We adjusted some sentences in the introduction hoping to make these points more understandable.

Point 2. The Materials and methods are not clearly presented. The whole part regarding the experimental design is very difficult to understand for the reader who has no knowledge of this issue. In the description of Figure 1, there are no explanations of all the abbreviations that appear, nor are the abbreviations UE and EU unambiguously used. When mentioning the formula for RA calculation, some reference according to which the formula was deduced has to be cited.

Thanks to the Reviewer for the advices. We corrected in Figure 1 UE in EU, as it stands for European Union. Abbreviations of Figure 1 were already reported in the legend (MA = Marketing authorization; a.i. = active ingredient; p.f. = pharmaceutical form). However, explanations of “EU”, “Art. 5, c.1 L. 94/98”, and “Art. 5, c.2 L. 94/98” were missing. Therefore, we added them in the legend. The reference for the formula for RA calculation was already reported in line124 (reference 18). For greater clearness, we added the reference also in the legend of Figure 2.

Point 3. Part of the text from lines 132 to 137 is not clear. A description of the work area is given, which would be much clearer if there was a scheme.

We thank for the advice. We added a planimetry of IOV-IRCCS’ UFA, which became Figure 3.

Point 4. When talking about microbiological controls (line 137), there should be a more detailed description of the procedures, sampling, number of samples taken, references should be cited for the tests that are mentioned, and references should be cited for the reagents used to perform the tests. The same applies to the equipment, for instance Fluke985 particle counter instrument, a better description should be given there.

The work within our centralized compounding unit is organized based on the Norms of Good Preparation (NBP) and Annex 1 of the European Union Guide to GMP (EU GGMP), and the ISO 4.8 guidelines according to EN/ISO 14644-1 (lines 131-140, 146-147, 157-158). Description of all controls, including microbiological ones, are detailed in these references and we found it redundant to slavishly report what was written in the aforementioned legislation. We understand that those who are not "into the field" and do not work in a centralized computing unit may find this part difficult to understand. Therefore, we enriched the description of the procedures reported in the section, as suggested.

Point 5. Table 1 contains a lot of data, but there is no reference to the literature according to which it was compiled. In line 168, “vial leftovers” are mentioned. It is not clear how many of these samples there were. Also, you mentioned testing with the LAL test kit, but it is not clear what exactly was tested. In my opinion, it is necessary to completely reorganize the methodological part of the work and clearly define each stage of the research and all methods.

We thank the Reviewer for the comments. Table 1 reports point-by-point the requirements of our QAS that must be fulfilled, and that must be considered in the self-assessment. As mentioned in the section “2.3. Design and application of the QAS”, QAS was designed using two reference texts: the Italian Ministry of Health guidelines, with particular focus on the Norms of Good Preparation (NBP) and Annex 1: “Manufacture of sterile products” (Good Manu-facturing Practice, GMP – European Commission). For clearness, we remind these two references in the legend of Table 1. As regards comments of lines 197-205, we compounded for each vial leftover an additional infusion bag. This was performed for the most prescribed a.i. and derived preparations in use in our hospital. TSB and TSM was performed for a pool of three different batches (of each preparation of each a.i.), while LAL test was performed according to manufacturer’s instructions for every single sample (therefore, not in pool). We added some information in this part to clarify.

Point 6. In the Literature review part (lines 179 – 184), the keywords on the basis of which the literature search was conducted have to be defined. It should be also defined what timespan was taken into the search. Then it should be said what criteria were used for the judgment, that is, how it was assessed that you stated "high level of evidence"; based on what was it concluded?

We agree with this comment. A better detailed description of the literature research was added, as suggested.

Point 7. Figure 3 is very difficult to read and understand. Since the areas to which the results refer (Lab 4, Lab 6 and Lab 8) were not defined in the section on Materials and methods, it is difficult to understand the results. Also, the numbers listed in the “hood” columns are difficult to understand. What do they refer to? All this is due to the fact that there is no well-described legend accompanying the table. Appropriate explanations must be given for all of the abbreviations listed in that figure.

We adjusted the Figure 3, now numbered Figure 4, and implemented the legend. As in materials and methods we added, according to the Reviewer’s kind advice, IOV-IRCCS’ UFA planimetry, the numbering of laboratories and hoods is now understandable.

Point 8. In part 3.3. descriptions of the work space are given. These descriptions should be explained in the methodological part of the paper, and the results should only refer to what was found. The present description is confusing and incomprehensible.

We thank the Reviewer for the kind advice. We integrated this part with that reported in section 2.3 of Materials and Methods.

Point 9. In part 3.4. it is stated that a "systematic and critical review of the literature" was made. However, this does not belong to the section Results. All such descriptions should be defined in the chapter on experimental design and methods. Wherever it is stated that something has been done according to the literature, it is necessary to cite in parentheses the appropriate reference numbers on which these statements are based. Table 6 is too complicated and its content is hard to follow because the font is very small.

We clarify the literature review, as already above-mentioned and re-edited table 6 to be more readable. An explanation of abbreviations was reported in the legend. We cannot simplify the content of the table because it is an extract of an original document and tool exploited and consulted as such by pharmacists, technicians, nurses, compounders, apprentices and students. 

Reviewer 2 Report

In my opinion the manuscript addresses the significant challenges faced by hospital pharmacies. I appreciate the quality of presentation (detailed description of methodology and results, linguistic correctness). However, I would like to suggest several minor corrections listed below:

- Figure 2: If I understand the last of the tables correctly, there are missing values in cells A2, A4 and C2 (at first I supposed the cells are intentinally omitted, however A4 value is mentioned later at line 198).

- lines 203-208: It seems to me that the criteria for lowering the RA value have not been presented earlier within the manuscript. I suggest introducing them e.g. within Methodology section.

- line 210: the 2021 year is mentioned, while Figure 3 is for the year 2022.

Author Response

We sincerely thank the Reviewer for taking time to carefully review the manuscript and give constructive criticisms and valuable comments, which helped to revise and improve the manuscript. Here, the response point-by-point to comments and suggestions.

Point 1. Figure 2: If I understand the last of the tables correctly, there are missing values in cells A2, A4 and C2 (at first I supposed the cells are intentinally omitted, however A4 value is mentioned later at line 198). Lines 203-208: It seems to me that the criteria for lowering the RA value have not been presented earlier within the manuscript. I suggest introducing them e.g. within Methodology section.

Thanks to the Reviewer for the observations. The empty cells are not missing values. Therefore, to improve clarity of the table, we modified Fig. 2 adding a “/” to each empty cell. As regard the A = 4 value reported in line 198, the number is correct. We reported: “Therefore, A = 4 since in the case of compounding of antiblastic drugs and parenteral nutrition bags in which the calculations are carried out with specific management software, the calculated value can be subtracted by 1 unit up to the minimum value of 4.”. We think that maybe the sentence was not clear. Therefore, as suggested, we added the criteria for lowering this value in the Material and Methods section, while the sentence in the results were modified as follows: “ The initial value of A is 5. However, since compounding of anticancer drugs in our UFA is carried out by aid of specific management software, A = 4.“

Point 3. line 210: the 2021 year is mentioned, while Figure 3 is for the year 2022.

Thanks to the Reviewer for noticing. It is a typo, the year 2022 is correct, so we corrected the sentence in line 210.

Reviewer 3 Report

The aim of the author’s study was to develop a methodology to evaluate the safety and preparation of chemotherapeutics as related to the patient.  It aims to advance current methods for analysis and provide guidelines in preparing chemotherapeutics. This work is relevant as it acknowledges the preparatory pitfalls (what could occur if aseptic technique aren’t employed) of synthesizing/compounding chemotherapeutics to the patient. The references are appropiate.

Specific improvements:

Method:
- Can the authors clarify where “aseptic tissue culture technique” was employed? I would assume yes but this needs to be explicitly stated.

Figures and Tables:

-General consideration for figure legends. I encourage the legends to be expanded to include more information (abbreviated description of the method as appropriate). The legend should be centered at the bottom of the figures.

-General consideration for figures is that they need to be alphabetized or something similar if there is more than one component, which is the case with many of the figures.

-Figure 2- Please alphabetize the three tables and reference them more specifically in the legend.  
-Table 1 and 2- Please size appropriately and group to heading.

Figure 3- alphabetize and consider breaking up to expand the sizing due to small font.

Please reformat the figures so the legend and the figure are grouped. I would suggest a different formatting for Figure 3. 

Author Response

We sincerely thank the Reviewer for taking time to carefully review the manuscript and give constructive criticisms and valuable comments, which helped to revise and improve the manuscript. Here, the response point-by-point to comments and suggestions.

Point 1. Method: Can the authors clarify where “aseptic tissue culture technique” was employed? I would assume yes but this needs to be explicitly stated.

We apologize to the reviewer but the phrase “aseptic tissue culture technique” is not reported as it is in our manuscript. If the Reviewer refers to the aseptic conditions during drug compounding, or the culture media used and procedures for microbiological controls, they are all specified in the Materials and Methods, section 2.3. We implemented this part to better describe and clarify all the procedures.

Point 2. Figures and Tables: -General consideration for figure legends. I encourage the legends to be expanded to include more information (abbreviated description of the method as appropriate). The legend should be centered at the bottom of the figures. -General consideration for figures is that they need to be alphabetized or something similar if there is more than one component, which is the case with many of the figures. -Figure 2- Please alphabetize the three tables and reference them more specifically in the legend. -Table 1 and 2- Please size appropriately and group to heading. Figure 3- alphabetize and consider breaking up to expand the sizing due to small font. Please reformat the figures so the legend and the figure are grouped. I would suggest a different formatting for Figure 3. 

We thank the Reviewer for the kind suggestions. We revised all figures and tables, making them clearer and more readable, implementing legends and alphabetization.